# Identifying Characteristics of Guam's Extreme Rainfalls Prior to Climate Change Assessment

Myeong-Ho Yeo [1,*] , James Pangelinan [1] and Romina King [2]

1 Water and Environmental Research Institute of the Western Pacific, University of Guam, Mangilao, GU 96923, USA; jamespangelinan@triton.uog.edu
2 Micronesia Area Research Center, University of Guam, Mangilao, GU 96923, USA; roking@triton.uog.edu
* Correspondence: yeom@triton.uog.edu; Tel.: +1-671-735-2693

**Abstract:** Extreme rainfall and its consequential flooding account for a devastating amount of damage to the Pacific Islands. Having an improved understanding of extreme rainfall patterns can better inform stormwater managers about current and future flooding scenarios, so they can minimize potential damages and disruptions. In this study, the scaling invariant properties of annual maximum precipitations (AMPs) are used for describing the regional patterns of extreme rainfalls over Guam. AMPs are calculated at seven stations in Guam and exhibit distinct simple scaling behavior for two different time frames: (1) from 15 min to 45 min; and (2) from 45 min to 24 h. With these two different behaviors, the conventional estimation methods for sub-hourly durations overestimate the frequencies at a site in which breakpoints are clearly observed, while the proposed Scaling Generalized Extreme Value (GEV) method, based on the Scaling Three-NCM (S3NCM) method, provides comparable estimates. A new regional extreme rainfall analysis approach based on scaling exponents is introduced in this study. Results show distinct extreme rainfall patterns over Guam. Moreover, the numerical and graphical analyses identify that a tropical cyclone may increase daily AMPs by 3%, on average.

**Keywords:** extreme rainfalls; flood; temporal downscaling; Guam; climate change

## 1. Introduction

Intensity–duration–frequency (IDF) curves of rainfalls describe the relationship between rainfall intensity and durations and frequencies. The IDF curves have been widely used for the planning and design of various hydraulic structures in urban hydrology and water resource management [1,2].

With annual maximum precipitations (AMPs), the computational procedure of modeling the IDF relationship can be summarized as: (*i*) select the appropriate distribution, (*ii*) estimate parameters of the selected distribution for each duration, and (*iii*) estimate quantiles (or intensities) with respect to the desired return periods. Probability distribution models have been used to describe the distribution of extreme rainfalls. National Oceanic and Atmospheric Administration (*NOAA*) *Atlas 14 Volume 5* [1] conducts rainfall frequency analyses with a generalized extreme value (GEV) distribution for the selected Pacific Islands and L-moment parameter estimator of the GEV distribution parameters. The parameter estimator yields more reliable values for small and highly skewed data sets than the conventional method of moments and maximum likelihood methods. Although the *NOAA Atlas 14* provides a comprehensive understanding of the extreme rainfalls over the Pacific Islands, engineers and designers still have trouble developing the IDF curves, caused by restricted data availability and climate change conditions [3,4].

Information on the extreme rainfalls in fine temporal resolution is critical for stormwater management purposes because most flash floods happen in short durations (e.g., less than 3 h). Since conventional models are unable to figure out the extreme rainfall's behavior for different time scales due to lack of data, rainfall frequency analyses in the Pacific

Islands have been limited. Therefore, to address the shortfalls of these conventional models, formulating models that statistically and simultaneously match various properties of the extreme rainfall process at different levels of aggregations is necessary.

Globally, more intense extreme weather events, such as tropical cyclones, tropical depressions, floods, and droughts, are highly likely to be expected with climate change [5]. Guam, a small Pacific Island, will be impacted by climate change [6–10]. Of particular concern, increased heavy rainfall events will result in flooding, erosion, and run-off—affecting major roads, highly valued real estate, coral reefs, and shallow coastal ecosystems [6]. A comprehensive understanding about extreme rainfall characteristics is essential for keeping communities safe from extreme weather events and water insecurity [3,11,12]. Assessing the climate change impacts on extreme rainfall behaviors in Guam is, therefore, crucial for supporting the Guam Hazard Mitigation Program.

Moreover, in Guam, tropical depressions and typhoons usually bring heavy rainfall with an estimated contribution of 12% of the annual precipitation average [13]. In the Western Pacific, these tropical disturbances are more frequent and intense in the initial half of El Nino durations [6,14]. Despite the presence of such important influences by them, few studies have been conducted on how much they affect IDF curves and design storms.

In the context of the aforementioned issues, the main objective of this paper is to propose an alternative GEV parameter estimator, estimating AMPs' intensities for different time frames to identify the extreme rainfall behaviors in Guam. Furthermore, the proposed estimator can support Guam's Hazard Mitigation Plan by updating the current IDF curves in accordance with climate change scenarios. The remainder of this paper is organized as follows:

Section 2 describes the methodologies for the parameter estimation methods using scale invariant properties of AMPs for Generalized Extreme Value (GEV) distributions for shorter durations and the investigation of typhoon effects on IDF curves; Section 3 provides information about the data used in this study; Section 4 discusses the application of the results to management; and Section 5 summarizes the conclusions.

## 2. Methodology

### 2.1. Development of Temporal Downscaling Model

Generalized Extreme Value (GEV) distribution is commonly used for fitting mathematically extreme precipitation data [1,15,16]. The cumulative distribution function (CFD), $F(x)$, for the GEV distribution is given as

$$F(x) = exp\left[-\left(1 - \frac{\kappa(x - \xi)}{\alpha}\right)^{\frac{1}{\kappa}}\right] \qquad (\kappa \neq 0) \qquad (1)$$

where $\xi$, $\alpha$, and $\kappa$ are the location, scale, and shape parameter, respectively. The quantiles $(X_\tau)$ corresponding to a return period can be calculated by the inverse distribution function as follows

$$X_\tau = \xi + \frac{\alpha}{\kappa}\{1 - [-ln(p)]^k\} \qquad (2)$$

where $p$ is the exceedance probability of interest. In this study, the Cunnane plotting position was implemented to calculate the probability.

In this study, scale invariance properties of AMPs are implemented for estimating the parameters of GEV distributions of sub-daily or sub-hourly duration using daily AMPs [17–20]. Thus, this model is a temporal downscaling model. For a scaling process, the relationship between the Non-Central Moments (NCMs) of order $k$ and the variable $t$ can be written in a general form as follows:

$$\mu_k = E\{f^k(t)\} = t^{\beta(k)}a(k) \qquad (3)$$

in which $\alpha(k) = \{f^k(1)\}$ and $\beta(k) = k\beta$. If the exponent follows a linear function, in such a case the process is considered to have a 'simple scaling' condition. Hence, the scaling

behavior of extreme rainfall can be examined by the power–law relationship between the *k*-order NCMs and the *t*-durations. If extreme rainfall data exhibit the scaling properties, the log-linearity will be shown.

As a simple scaling process, it can be shown that the statistical properties of the GEV distribution for two different time scales *t* and $\lambda t$ are related as follows:

$$\kappa(\lambda t) = \kappa(t) \tag{4}$$

$$\alpha(\lambda t) = \lambda^{\beta}\alpha(t) \tag{5}$$

$$\xi(\lambda t) = \lambda^{\beta}\xi(t) \tag{6}$$

$$X_T(\lambda t) = \lambda^{\beta}X_T(t) \tag{7}$$

Based on these relationships, it is possible to derive the statistical properties of sub-daily AMPs using the properties of daily AMPs. Hereafter the temporal downscaling model is called the Scaling-GEV model. The three temporal downscaling methods, which are the Scaling-Lmoment method (SLmom), the Scaling One-NCM method (S1NCM), and the Scaling Three-NCM method (S3NCM), can be derived through the Equations (3)–(7). Detailed description of the temporal downscaling methods can be found in Yeo, Nguyen, and Kpodonu [16]. In this study, the three temporal downscaling methods are applied to estimate sub-daily and sub-hourly AMP quantiles using the measured daily AMPs. Comparison analyses are carried out for determining the best temporal downscaling method for Guam's extreme rainfall process.

## 2.2. Investigation of the Effect of Two Distinct Storm Systems on IDF Curves Estimation

When determining the multiple or simple scaling properties of AMP series, break-points have been observed in the middle of the NCM plots [4,18,21]. In general, the observed breakpoints are located between 30 min and 1 h. Since the presence of a break-point may imply the transition of rainfall dynamics from a small-scale storm system to that of a large-scale system, such as tropical depressions or typhoons, it is likely to influence the IDF estimates for the sub-hourly durations. However, *NOAA Atlas 14* [1] constructs IDF curves using scaling factors and available AMP series' at 1 h to estimate AMP frequencies at 5 to 30 min. In this study, the comparison study, therefore, is carried out via the estimated AMPs by two log-linear regression models, the scaling factors, and the Scaling-GEV model based on the S3NCM parameter estimator with the observed AMPs. Scaling factors are 0.50 and 0.74 for 15 min and 30 min durations, respectively. In addition, the first log-linear regression model uses six AMP series (1 h, 2 h, 4 h, 6 h, 12 h, and 24 h), while the second model uses only two AMP series (1 h and 24 h) for building regression models. The procedure can be described as the following.

(a)     To build log-linear regression models for each return period using the six- or two-AMP  series.
(b)     To estimate the three AMP series (15 min, 30 min).
(c)     To compare the estimated AMP series to the observed AMP series.

## 2.3. Investigation of Typhoon Effects on IDF Curves

Guam is located in the path of typhoons, which are matured tropical cyclones, resulting in damage, both in terms of loss of lives and economic costs [22,23]. Guam, on average, receives one typhoon passing within 60 nautical miles (nm) per year [1,24]. However, with climate change it is anticipated that there may be stronger/fewer tropical storms and typhoons [25]. Flooding is not limited to a typhoon event but may also result from short storm events such as squalls. It is, therefore, essential to determine the extent typhoons affect IDF curves for current and future periods, especially when calculating stormwater estimates. In this study, frequency analyses and applications of temporal downscaling models are carried out using two sets of AMPs: one is prepared using the whole 15 min

rainfall records, and the other is prepared by using the 15 min records excluding typhoons within 60 nm from Guam.

In this study, the characteristics of Guam's extreme rainfall are identified using three major components: the temporal downscaling method, the effects of storm system transition on IDF curves, and the typhoon effects on IDF curves. Figure 1 shows the scheme of the presented study procedure.

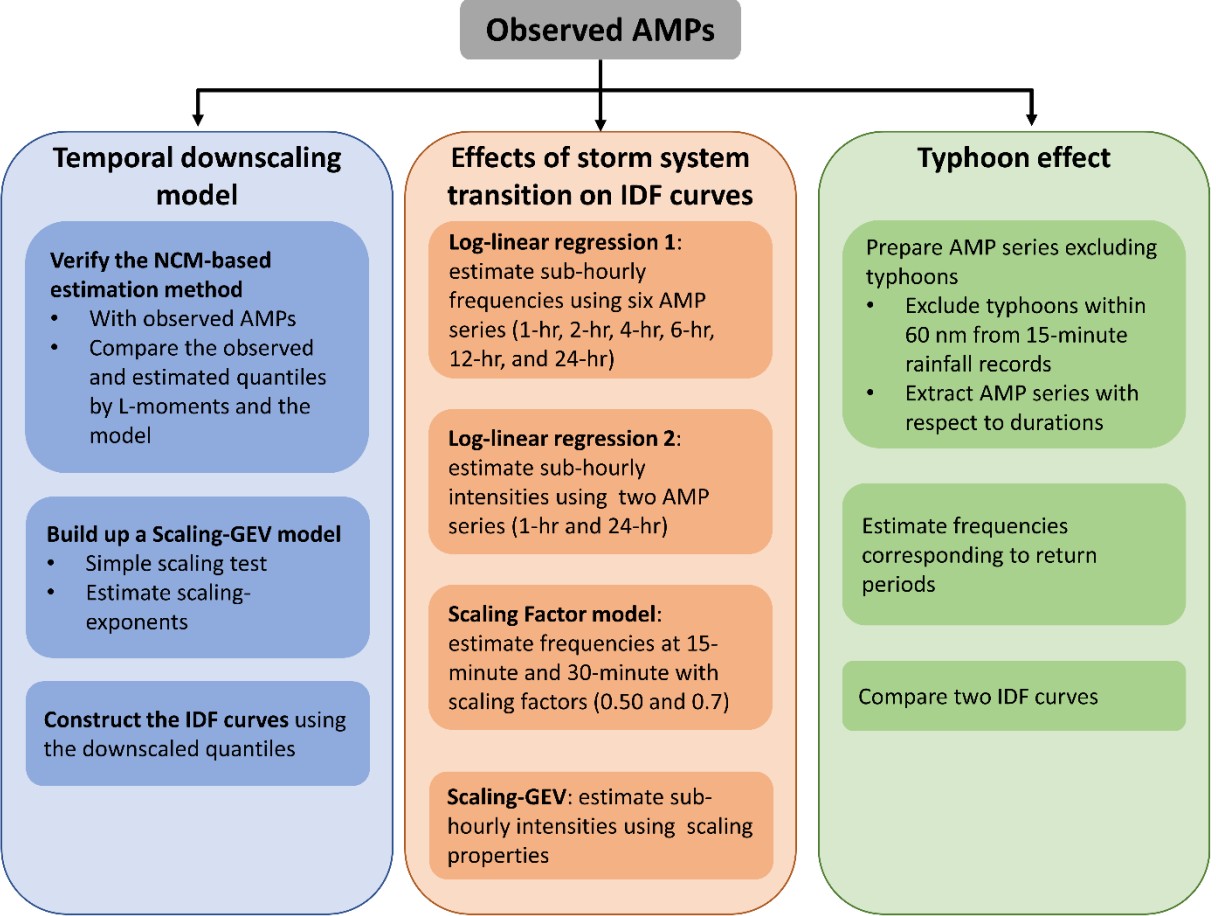

**Figure 1.** Scheme of this study.

## 3. Study Area and Data

Guam (located at 13°28′ N, 144°45′ E) is the largest and southernmost island of the Mariana Islands (Figure 1). Guam is 50 km long and 14 km wide (at its widest point) or approximately 550 km². The Adelup–Pago major fault divides Guam into a northern limestone plateau and a southern volcanic province. The northern half is a broad uneven limestone plateau with precipitous coastal cliffs standing 60–180 m above sea level, while the southern portion is a dissected volcanic upland with the cliff standing 70 m above sea level [26,27]. The interior northern plateau is predominantly composed of Miocene–Pliocene Barrigada Limestone [26]. The Pliocene–Pleistocene Mariana Limestone, composed of a reef and lagoonal deposit, forms the rim of northern Guam. Due to the porous nature of limestone, there is no surface stream flow on the northern limestone plateau. In contrast, there are a number of streams and rivers in the southern volcanic portion of the island (See Figure 2). The general climate of Guam is tropical, humid, and hot throughout the year. The island experiences two main seasons: dry (from December to May) and rainy (from July to November) seasons. Average annual rainfall in the dry season is estimated as 80.34 cm/year and 178.10 cm/year in the rainy season [13]. Two dominant storm systems contributing to the rain characteristics are small-scale storms (e.g., thunderstorms and squalls) and large-scale storms (e.g., tropical depressions and

typhoons). About 80% of annual peak discharges induced by the combination of small- and large-scale storms are observed for the rainy season [22].

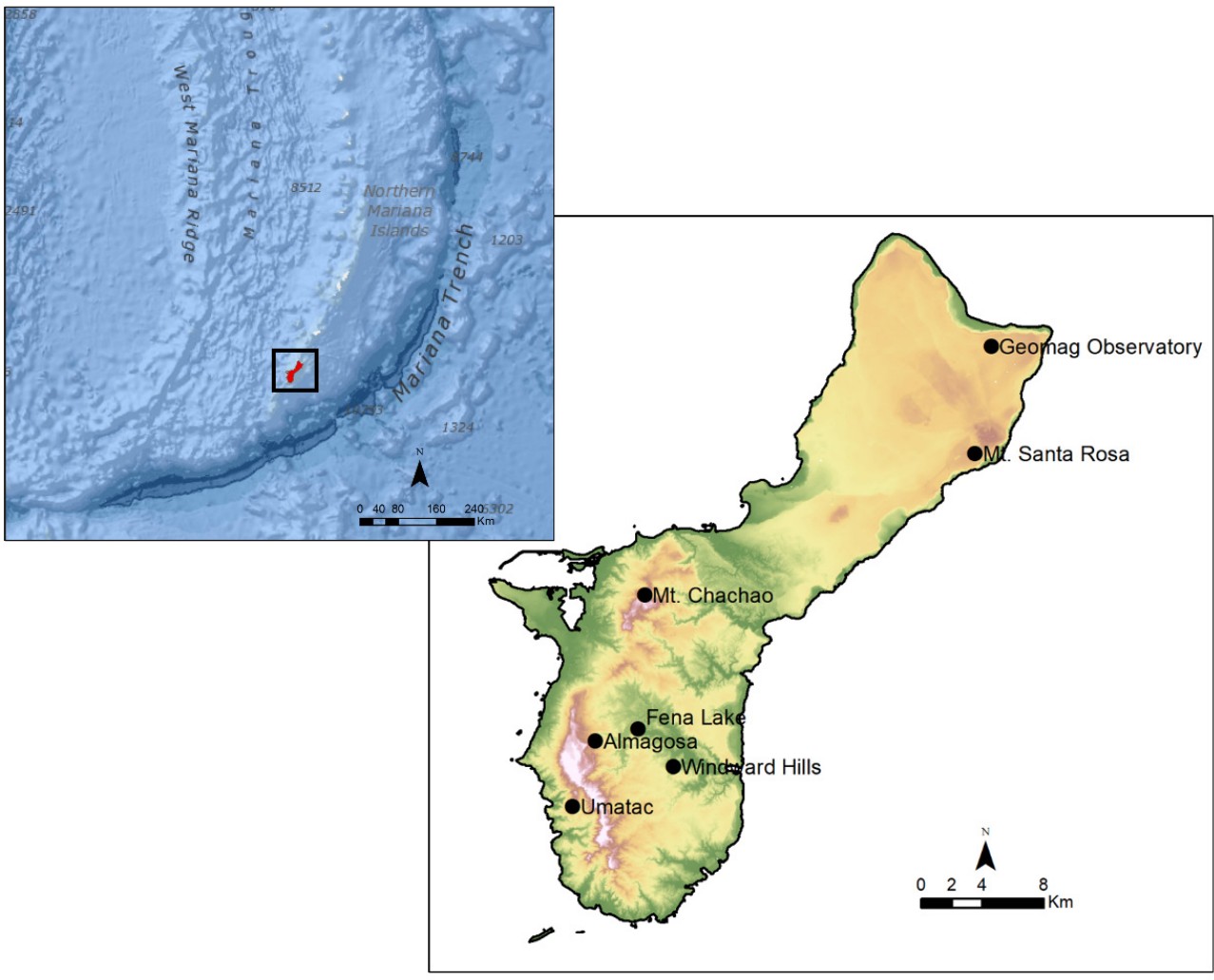

**Figure 2.** Geographical information on the study area. Upper map shows the location of Guam in the Mariana region, and the bottom map shows the digital elevation map. Black points denote the rain gages used in this study. Sources for World Ocean Reference and Bases in the upper map are Esri, GEBCO, NOAA, National Geographic, Garmin, HERE, Geonames.org, and other contributors.

To illustrate the application of the proposed temporal downscaling approach, case studies are conducted using an on-site AMP series, available from seven rain gage stations in Guam (Table 1 and Figure 2). Although the *NOAA Atlas 14 Volume 5* [1] and the *CNMI and Guam Stormwater Management Manual* [28] provide results of the frequency analyses, there is no historical AMP series for further impact studies, such as feasibility tests or climate change assessments. In this study, historical AMP series are calculated from active rain gage stations, which measure rainfall intensities in 15 min intervals and are operated by the United States Geological Survey (USGS) Pacific Islands Water Science Center (http://hi.water.usgs.gov/studies/project_waterdata.htm, accessed on 14 April 2022) by extracting the maximum intensity for each duration. Thus, the constructed AMP series ranges from 15 min to 1 day (i.e., 15 min, 30 min, 45 min, 1 h, 2 h, 3 h, 6 h, 12 h, and 24 h). Moreover, we are preparing an additional AMP data set to investigate how much typhoons influence the IDF curves. The second AMP series is calculated using the same procedure from the USGS data set that excludes rainfall records when typhoons arrive within 60 nautical miles (nm) of Guam's coast. For example, if a typhoon hits Guam

on 4 July, the second AMP series is constructed without rainfall records from 2 July to 6 July so as to remove the typhoon effects. The typhoon records provided by the National Weather Service Weather Forecast Office Guam in Table 2 are eliminated from the second AMP series.

**Table 1.** Geographical information of the selected 7 rain stations.

| Site | Site IDs | Latitude | Longitude | Length |
|---|---|---|---|---|
| Almagosa Rain Gage | GA1 | 13.35292 | 144.68314 | 2007–2021 |
| Fena Rain Gage near Reservoir Pump Station | GA2 | 13.36025 | 144.70889 | 2007–2021 |
| Geomag Observatory | GA3 | 13.58820 | 144.92020 | 2012–2021 |
| Mt. Chachao Rain Gage | GA4 | 13.43947 | 144.71217 | 2010–2021 |
| Mt. Santa Rosa, Yigo | GA5 | 13.52460 | 144.91090 | 2001–2021 |
| Umatac Rain Gage, Umatac | GA6 | 13.31390 | 144.66980 | 2007–2021 |
| Windward Hills, Yona | GA7 | 13.33830 | 144.73020 | 2007–2021 |

**Table 2.** Information about typhoons reaching to within 60 nm of Guam.

| Name of Typhoon | Date (yyyy-mm-dd) | Distance from Guam (km) | Intensity (m/s) | Max Intensity within Radius (m/s) |
|---|---|---|---|---|
| Krosa | 2001-10-03 | 82.8 | 12.9 | 15.4 |
| Chataan | 2002-07-05 | 50.7 | 48.9 | 48.9 |
| Pongsona | 2002-12-08 | 33.7 | 66.9 | 66.9 |
| Krovanh | 2003-08-17 | 28.3 | 12.9 | 12.9 |
| Maemi | 2003-09-05 | 21.9 | 15.4 | 15.4 |
| Maeri | 2004-09-20 | 99.5 | 12.9 | 15.4 |
| Tokage | 2004-10-12 | 98.2 | 18.0 | 18.0 |
| Talim | 2005-08-25 | 75.0 | 10.3 | 10.3 |
| Saomai | 2006-08-05 | 57.0 | 23.2 | 23.2 |
| Nuri | 2008-08-16 | 51.7 | 7.7 | 7.7 |
| Dolphin | 2008-12-11 | 75.0 | 15.4 | 15.4 |
| Mirinae | 2009-10-27 | 85.0 | 20.6 | 20.6 |
| Malou | 2010-08-31 | 53.5 | 7.7 | 7.7 |
| Megi | 2010-10-11 | 85.7 | 7.7 | 7.7 |
| Sanvu | 2012-05-22 | 87.2 | 20.6 | 20.6 |
| Wipha | 2013-10-10 | 32.4 | 10.3 | 10.3 |
| Francisco | 2013-10-16 | 97.0 | 15.4 | 15.4 |
| Rammasun | 2014-07-12 | 35.4 | 15.4 | 15.4 |
| Halong | 2014-07-30 | 82.8 | 28.3 | 28.3 |
| Bavi | 2015-03-15 | 11.3 | 20.6 | 20.6 |
| Dolphin | 2015-05-15 | 79.5 | 51.4 | 51.4 |
| Chan-Hom | 2015-07-05 | 45.4 | 23.2 | 23.2 |
| Three | 2016-07-13 | 73.5 | 7.7 | 7.7 |
| Fourteen | 2016-08-23 | 85.9 | 20.6 | 20.6 |
| Meranti | 2016-09-07 | 103.0 | 7.7 | 7.7 |
| Malakas | 2016-09-11 | 72.2 | 15.4 | 15.4 |
| Chaba | 2016-09-28 | 90.2 | 15.4 | 15.4 |
| Maria | 2018-07-04 | 20.7 | 28.3 | 30.9 |
| Soulik | 2018-08-15 | 100.6 | 15.4 | 15.4 |
| Mahgkhut | 2018-09-10 | 90.2 | 51.4 | 51.4 |
| Trami | 2018-09-20 | 51.5 | 12.9 | 12.9 |

## 4. Results and Discussion

### 4.1. Scaling-GEV Model

The comparison study is implemented for verifying the feasibility of the proposed parameter estimator using the first three NCMs. Once getting three parameters of the GEV distribution for each duration using both the L-moment-based and the proposed estimators, AMP quantiles are calculated with respect to exceedance probability. Figure 3 shows the

observed and estimated quantiles for durations of 15 min and 1 h for two representative stations (Almagosa and Mt. Chachao, respectively). It is found that the quantiles estimated by the proposed method fit closely to the quantiles generated by the conventional method and observed data.

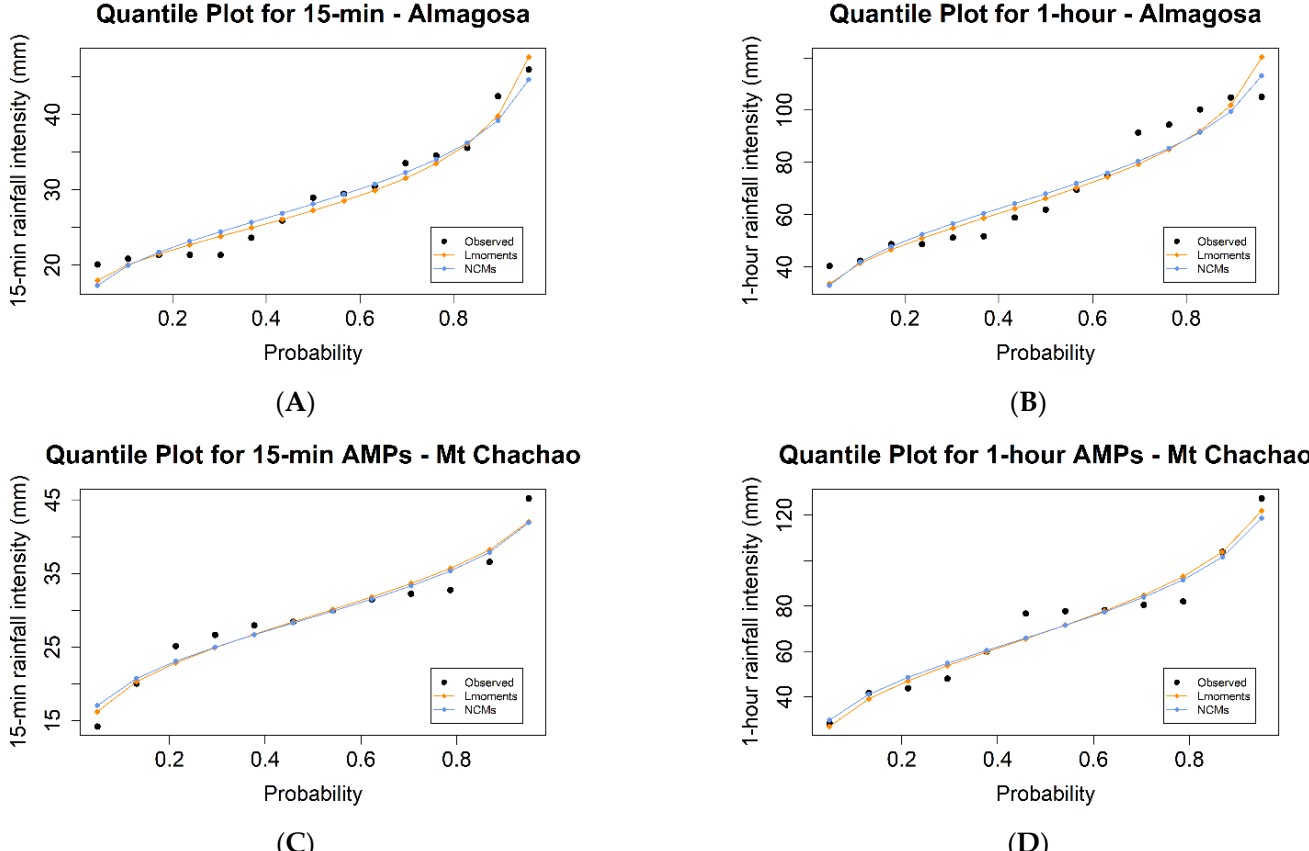

**Figure 3.** Quantiles estimated by conventional parameter estimation method (L-moments) and the proposed NCMs. (**A**,**B**) are quantiles plot for 15-min and 1-h durations for Almagosa, and (**C**,**D**) are those for Mt. Chachao.

To examine the temporal scaling properties of the AMP series, graphical analyses are carried out for all seven stations using the first three NCMs. Figure 4 shows the scaling relationship with respect to all durations. The log-linearity of NCMs illustrates two distinct regimes: from 15 min to 45 min and from 45 min to daily. The breakpoints for Guam's rain gages are observed at 45 min. In addition, the linearity of scaling exponents, $\beta(k)$, which are slopes of the log-linear lines shown in Figure 4, against the order of NCMs of AMPs for all stations, as shown in Figure 5, has indicated that the AMPs for all stations can be described by a simple scaling model. It is therefore possible to build up temporal downscaling models using the Equations (4)–(7). Although Alamosa (A) and Mr. Chachao station (D) in Figures 4 and 5 show the largest differences in the values of the scaling exponents for the shorter and longer durations, the values of the scaling exponents at Geomag and Mt. Santa Rosa stations are shown as parallel in Figure 5C,D. The statistical meaning of the scaling exponent is the ratio of average rainfall intensity to unit duration. The steeper slope or the higher value of the exponent indicates a greater increase in rainfall intensity during the specific duration. Therefore, the former two stations are likely to be vulnerable to a small-scale storm system, but the latter two stations are subject to tropical depressions or typhoons.

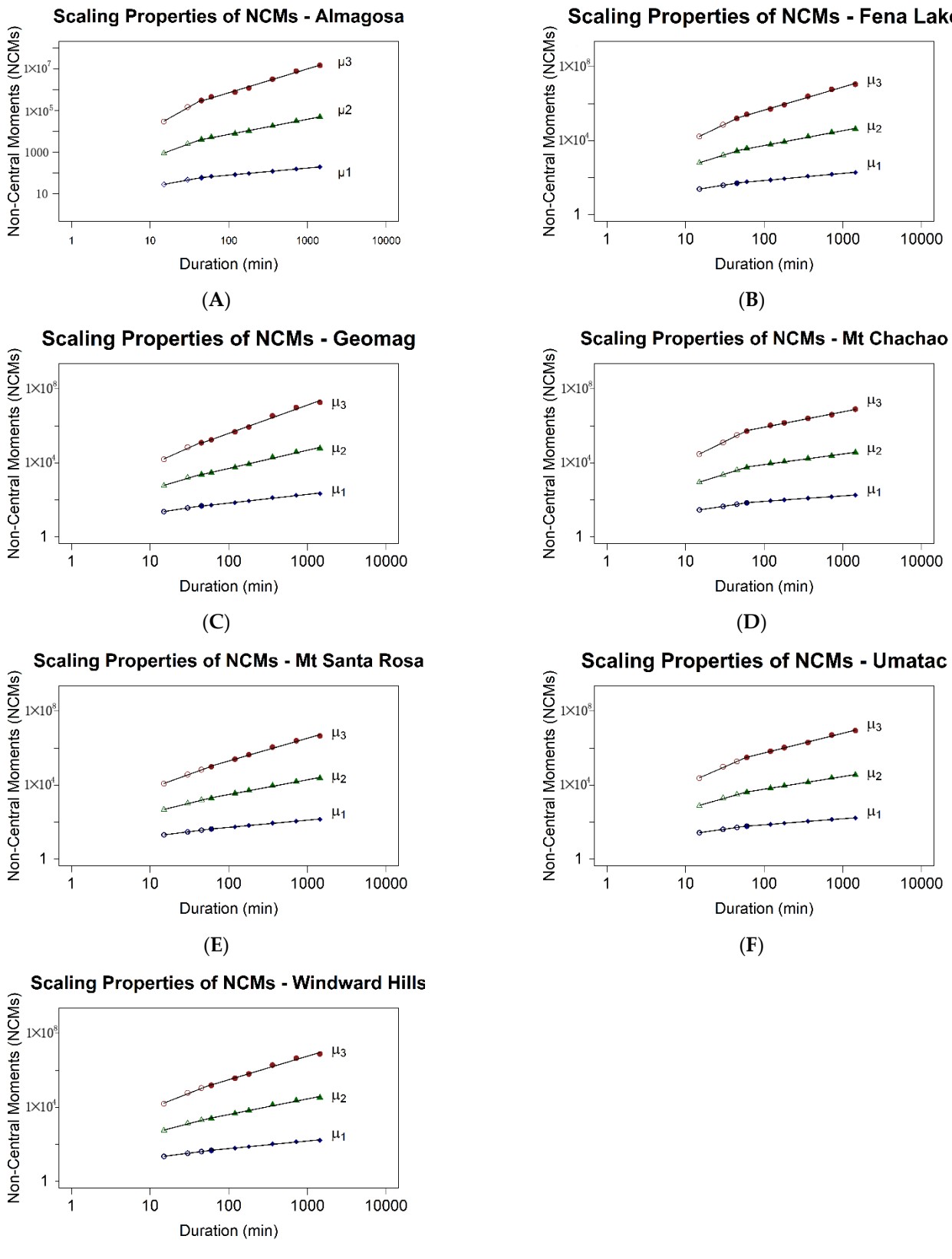

**Figure 4.** Scaling properties of AMPs. (**A**) is the NCM plot for Almagosa, (**B**) for Fena Lake, (**C**) for Geomag, (**D**) Mt. Chachao, (**E**) for Mt. Santa Rosa, (**F**) for Umatac, and (**G**) for Windward Hills, respectively.

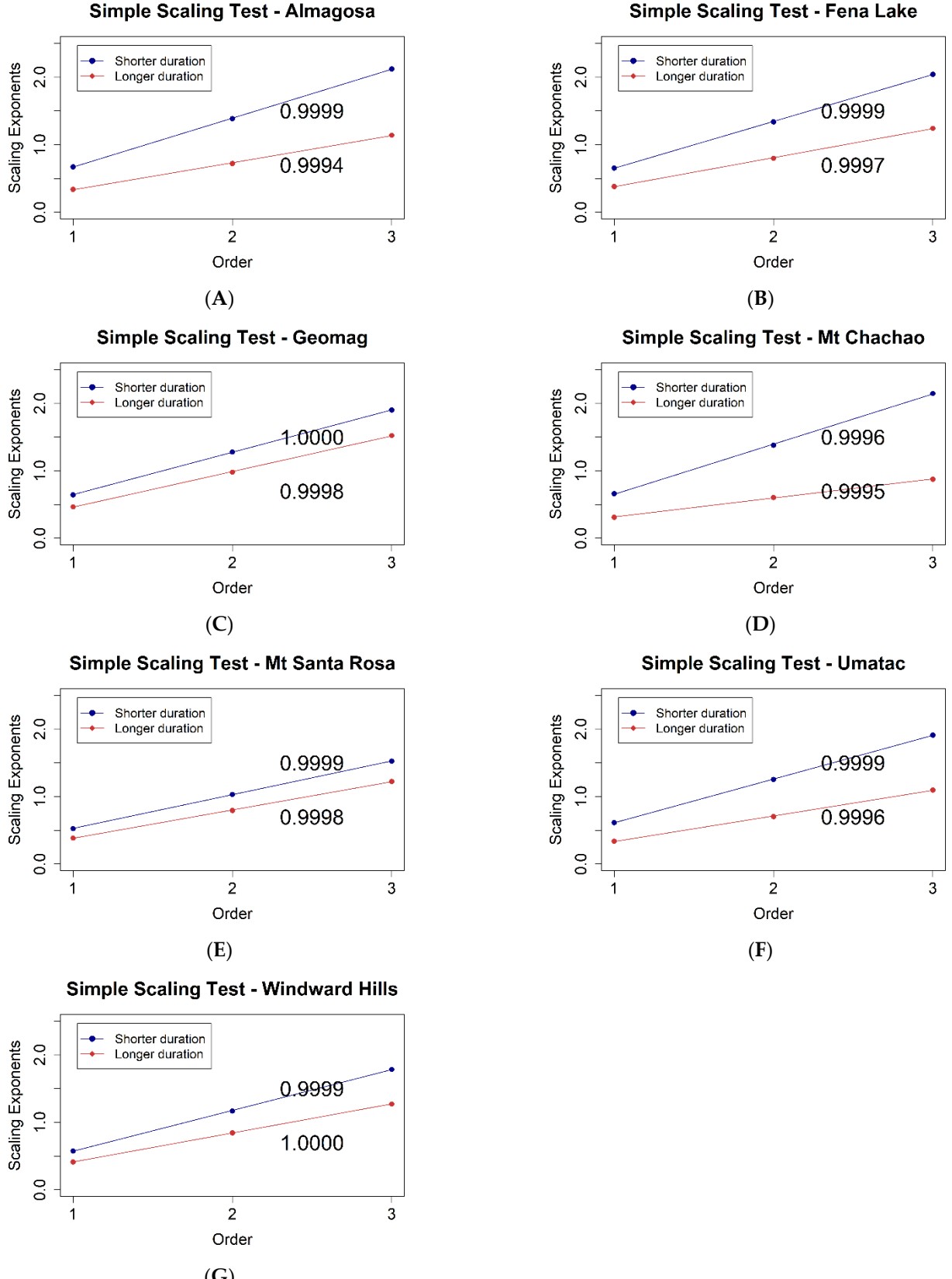

**Figure 5.** Test of Simple scaling. (**A**) is a plot the scaling exponent against order of NCM of AMPs for Almagosa, (**B**) for Fena Lake, (**C**) for Geomag, (**D**) Mt. Chachao, (**E**) for Mt. Santa Rosa, (**F**) for Umatac, and (**G**) for Windward Hills, respectively.

To investigate the effects of the transition of storm systems, the comparison study is carried out using various estimation methods that are two log-linear regression methods, the Scaling Factor method, and the proposed Scaling-GEV based on S3NCM method. Mt. Chachao (GA4) and Mt. Santa Rosa (GA5) are selected for the comparison study because GA4 shows two regimes (Figure 4D) and GA5 displays one regime (Figure 4E). In Figure 6, black dots represent the observed quantiles, sky-blue diamonds do the estimates by Scaling-GEV, orange triangles by log-linear regression using the observed six AMP series, green circles by log-linear regression using two AMP points, and dark blue squares by Scaling Factor method. The Scaling Factor and Scaling-GEV methods provide very close estimates to the observed at the GA5 station. However, the largest difference between the observed and estimated by two regression methods and the Scaling Factor method is shown in the IDF curves for 15 min duration at GA4, but the estimates for 45 min duration are very similar to the observed. The estimated frequencies by Scaling-GEV are very close to the observed regardless of time durations. Because the presence of breakpoint implies the transition of storm system, it is likely to overestimate or underestimate the quantiles for short durations using the quantiles for longer durations at a station in which the breakpoint exists. Results shows the proposed Scaling-GEV method provides robust results regardless of the storm transition.

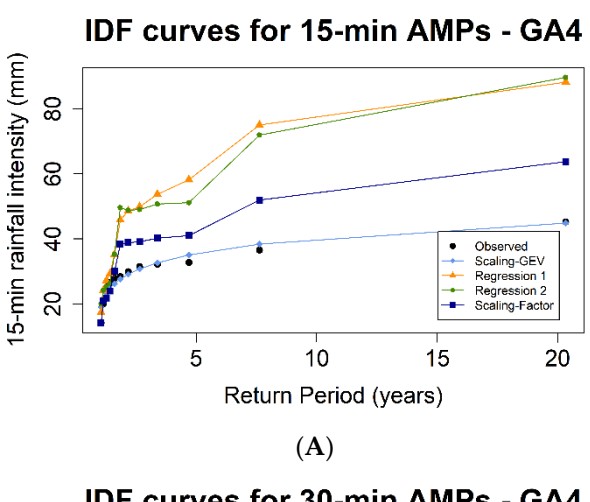

(**A**)

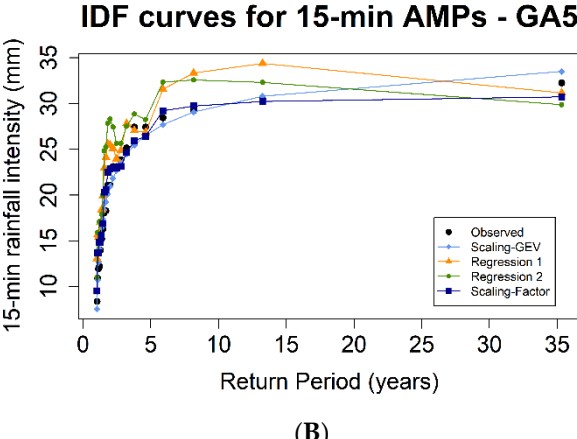

(**B**)

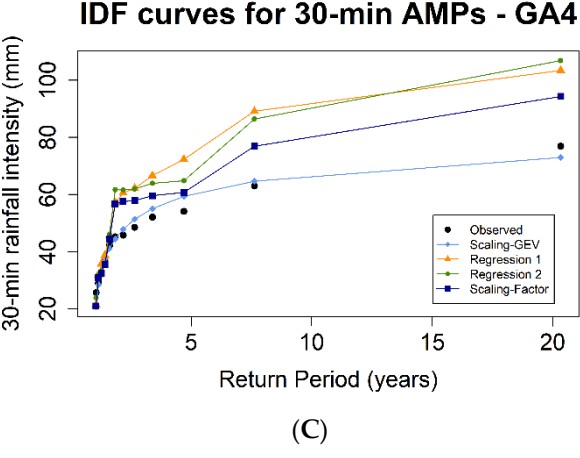

(**C**)

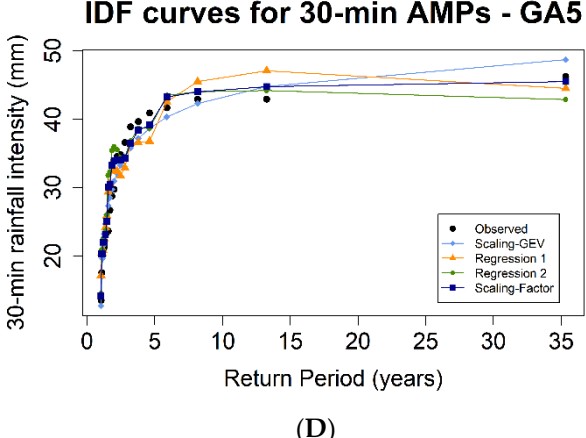

(**D**)

**Figure 6.** *Cont.*

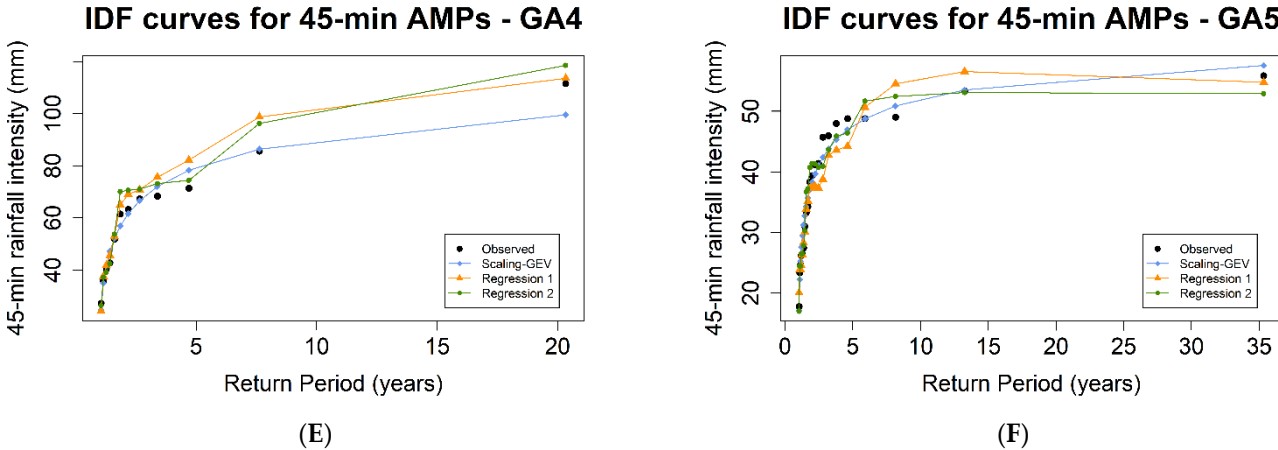

**Figure 6.** IDF curves for comparing the estimation method of sub-hourly frequencies. Two stations (Mt Chachao and Mt Santo Rosa) are selected. Breakpoints are clearly observed at Mt Chachao (GA4) but not clear at Mt. Santa Rosa (GA5). (**A**,**C**,**E**) are IDF curves for the 15-min, 30-min, and 45-min durations for Mt. Chachao station, and (**B**,**D**,**F**) are those for Mt. Santa Rosa, respectively.

To identify the regional heavy rainfall characteristics, the inverse-distance weight (IDW) method as a geospatial interpolation is applied to the estimated scaling exponents with coordinates of rain gages for regional analyses. Figure 7 shows regional scaling exponent maps for the short duration (A) and the long duration (B), respectively. Again, the northern portion of Guam is a broad uneven limestone plateau, while the southern portion is a volcanic upland. The orographic features, which are characterized by Mount Alutom and Mount Lam Lam, located in Southern Guam, induce a more intense rainfall amount with a short duration as shown in Figure 6A. On the contrary, the northern portion receives more rainfall for a longer duration from 45 min to 24 h. It implies that the southern portion, having the high values of slopes, is more likely to be vulnerable to flash floods induced by intensive storm rainfall within a very short duration.

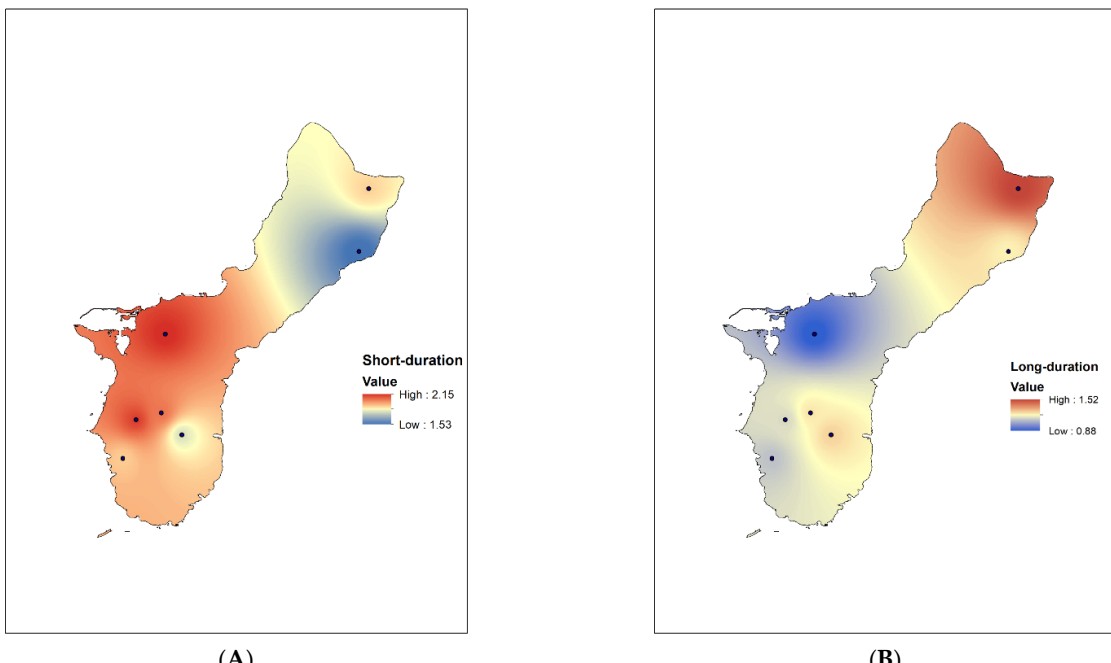

**Figure 7.** Interpolated scaling exponents for both short duration (**A**) and long duration (**B**).

The three proposed temporal downscaling methods (SLmom, S1NCM, and S3NCM) are used to construct IDF curves for the current period. In Figure 8, black dots represent the observed rainfall intensities, orange diamonds for the estimated values by SLmom, green triangles for those by S1NCM, and blue circles for those by S3NCM. Overall, the S3NCM method estimates accurately extreme quantiles. Hence, for the temporal downscaling modelling, the S3NCM method will be used for estimating the GEV parameters for sub-daily and/or sub-hourly durations. Furthermore, we plan to incorporate the developed downscaling models using S3NCM in the following climate change adaptation studies.

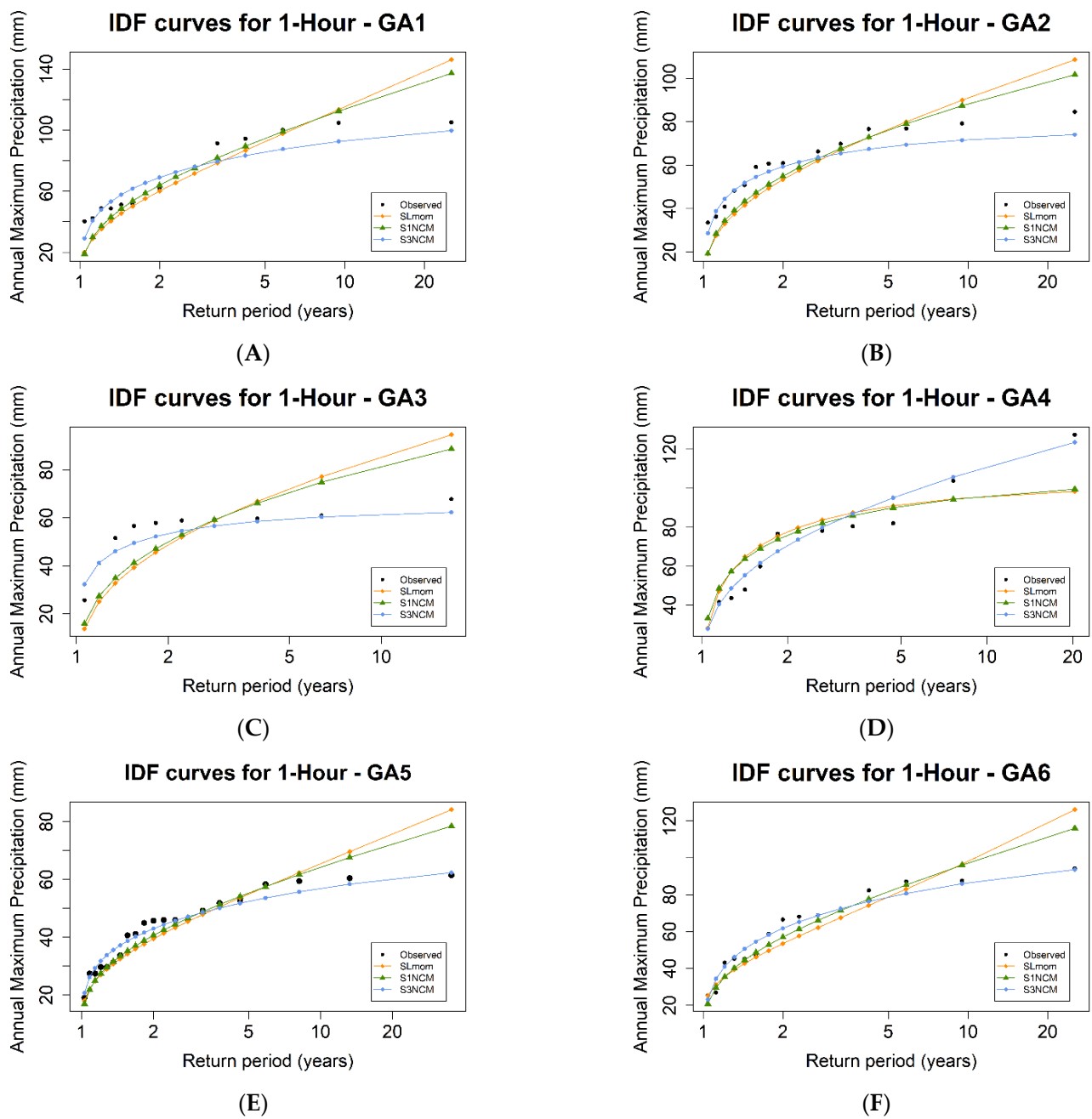

**Figure 8.** *Cont.*

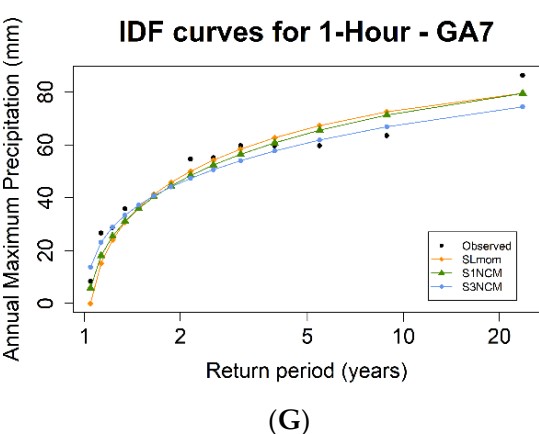

(**G**)

**Figure 8.** Estimated IDF curves using three temporal downscaling approaches. The 1-h IDF curves for Almagosa (**A**), Fena Lake (**B**), Geomag (**C**), Mt. Chachao (**D**), Mt. Santa Rosa (**E**), Umatac (**F**), and Windward Hills (**G**), respectively.

*4.2. Investigation of Typhoon Effects on IDF Curves*

In this study, numerical analyses are carried out using the relative-root-mean-square error (RRMSE) to quantify the typhoon effect on extreme rainfall intensities. The RRMSE is expressed by

$$\text{RRMSE} = \sqrt{\frac{1}{n} \sum_{i=1}^{n} \left( \frac{\left(X_i^{Raw} - X_i^{60\text{nm}}\right)^2}{X_i^{Raw}} \right)^2} \tag{8}$$

where $X_i^{Raw}$ and $X_i^{60\text{nm}}$ are the estimated quantiles using the raw AMPs records and the AMPs excluding typhoon records passing within 60 nm from Guam, respectively. The smaller the value of RRMSE, the lower the effect of typhoons on the annual extreme series. In Table 3, bold letters indicate the largest value for each duration. The site having a bold letter is the location of concern with relation to typhoon effects. Numerical assessment results show that Geomag, which is located in northern Guam, has the largest rainfall amount among the local sites when a typhoon is passing.

**Table 3.** RRMSE Values.

| Durations | Almagosa | Fena Lake | Geomag | Mt Chachao | Mt Santa Rosa | Umatac | Windward Hills |
|---|---|---|---|---|---|---|---|
| **15 min** | 0.51 | 0.80 | **1.89** | 0.00 | 0.00 | 0.00 | 0.55 |
| **30 min** | 0.00 | 0.42 | **3.42** | 0.00 | 0.00 | 0.00 | 0.73 |
| **45 min** | 0.00 | 0.16 | **1.88** | 0.00 | 0.10 | 0.20 | 0.75 |
| **1 h** | 0.00 | 0.20 | **1.42** | 0.23 | 0.46 | 0.95 | 0.36 |
| **2 h** | 0.33 | 0.52 | **2.82** | 1.01 | 1.36 | 0.67 | 0.58 |
| **3 h** | 0.61 | 0.76 | **4.13** | 1.63 | 1.71 | 1.25 | 2.24 |
| **6 h** | 1.66 | 1.49 | **8.75** | 2.33 | 2.73 | 1.43 | 3.70 |
| **12 h** | 2.13 | 2.67 | **11.91** | 2.22 | 3.35 | 1.66 | 3.80 |
| **24 h** | 2.78 | 2.91 | **7.77** | 1.69 | 2.66 | 1.49 | 4.39 |

In addition to the numerical assessment, Figure 9 shows the graphical IDF curves constructed by the conventional L-moment methods and the two sets of observed AMPs. Blue solid lines denote the IDF curves estimated by the raw AMPs, while red-dash lines denote those by the AMPs without typhoon records. Like the numerical assessment results, the biggest gaps between two AMPs are observed at Geomag (Figure 9C). It implies that the northern portion of Guam receives heavier rainfall amounts during typhoon conditions than the remaining portion of Guam.

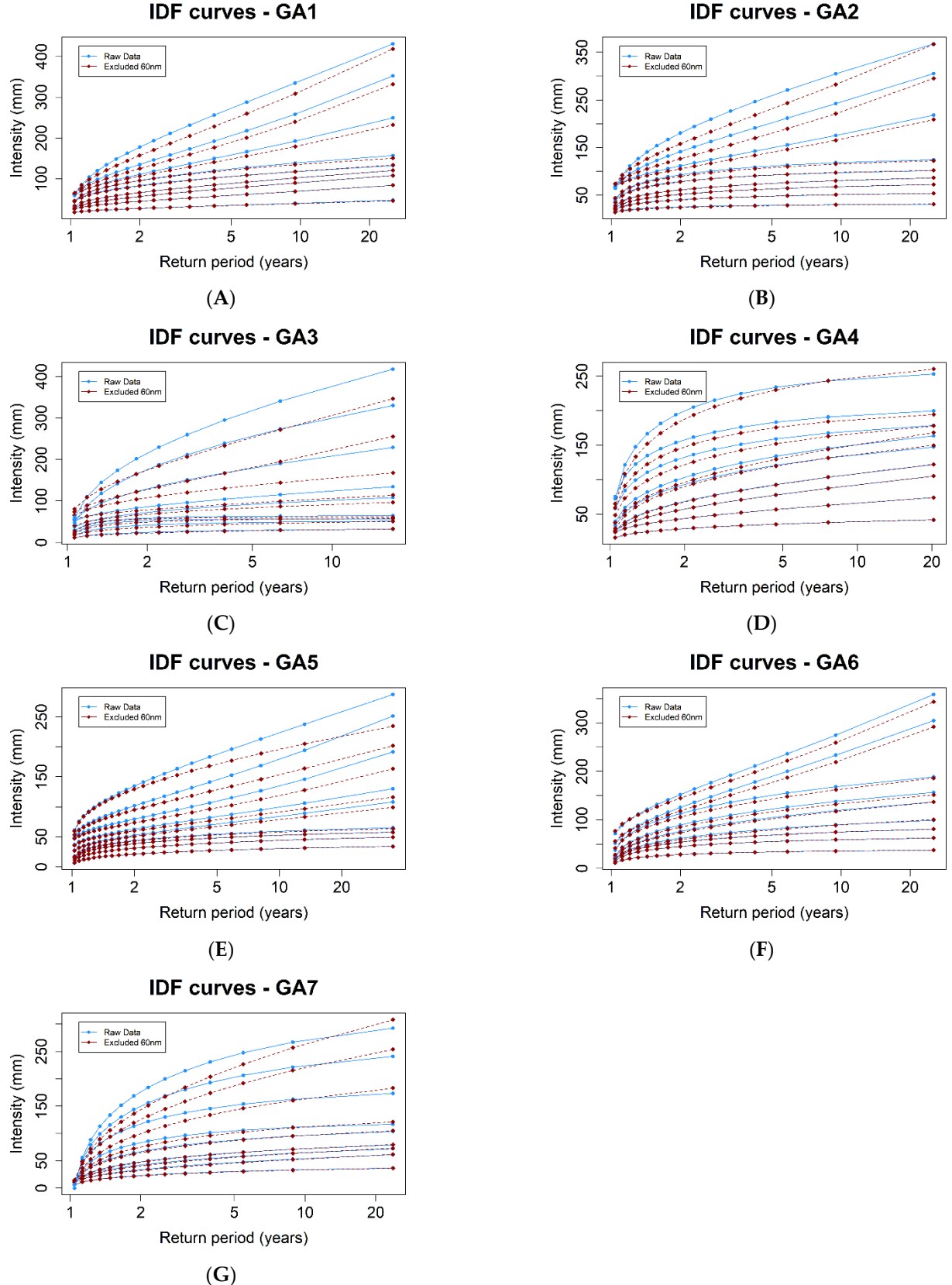

**Figure 9.** Comparison of IDF curves for accounting for typhoon effects on IDF curves. The estimated IDF curves for Almagosa (**A**), Fena Lake (**B**), Geomag (**C**), Mt. Chachao (**D**), Mt. Santa Rosa (**E**), Umatac (**F**), and Windward Hills (**G**), respectively.

## 5. Conclusions

Extreme rainfall and its consequential flooding account for a devastating amount of damage to the Pacific Islands. Moreover, these extreme weather events have been observed in increasing trends over the globe [29]. Having an improved understanding of extreme rainfall patterns can better inform stormwater managers about current and future flooding scenarios, so they can minimize potential damages and disruptions.

In this study, we apply three temporal downscaling methods (SLmom, S1NCM, and S3NCM) associated with the GEV distribution towards the historical AMP series from seven rain gage stations in Guam for the recent period. AMPs at these stations exhibit simple scaling behavior within two different time frames: (1) from 15 min to 45 min; and (2) from 45 min to 24 h. Based on the simple scaling property, three scaling GEV distribution models are developed for estimating sub-daily and sub-hourly AMPs using only available daily AMPs. The comparison study shows the temporal downscaling model using the first three NCMs provides the most accurate quantiles for Guam's stations.

The comparison study is carried out to investigate the effects of storm system transition on the IDF curves at short durations. Two log-linear regression methods, the Scaling Factor method, and the proposed Scaling-GEV method are used to estimate frequency estimates at 15 min and 30 min durations for two representative stations (Mt. Chachao and Mt. Santa Rosa). Results have indicated that the conventional methods (Scaling Factor and two log-linear methods) overestimate the frequencies at a site in which breakpoints are clearly observed. However, the proposed Scaling-GEV method, based on the S3NCM, provides comparable estimates regardless of the presence or absence of breakpoint.

Consequently, two remarkable advantages of the proposed Scaling-GEV model can be recognized. As a parsimonious parameter estimator, the model allows a modeler to estimate the frequencies for short durations using only the daily AMP series. Another benefit is to provide robust estimates regardless of the presence or absence of breakpoint in the NCMs' fields.

Furthermore, this study offers a new regional method using the geospatial interpolation of the scaling exponents. Because the exponent is the average ratio of extreme rainfall intensity to unit duration, the value would be used for comparing the local characteristics of extreme rainfalls. From the scaling exponent maps, it is found that southern Guam receives more intense rainfall within a short duration, relative to the orographic features, than the northern portion. Hence, the catchments and watersheds are identified as possible flood-prone areas during flash flood events.

In this study, the typhoon effect on extreme rainfalls is numerically/graphically justified. Because major tropical storms and typhoons have passed through the Mariana region, acknowledgement of the impacts of typhoons on the IDF curves is helpful for engineers to design stormwater management and sewer systems. Two sets of IDF curves using the original AMP series and the updated AMP series excluding the typhoon records are compared. On average, typhoons lead to an increase in daily IDF curves by about 3%. Although typhoons are one of the most significant concerns to Guam, floods are not just limited to typhoons, but also short-duration thunderstorms.

For future research, we will further refine the scaling exponent maps for the regional frequency analyses associated with ungauged locations and couple with updated climate change scenarios to update the current IDF curves.

**Author Contributions:** Conceptualization, M.-H.Y.; methodology, M.-H.Y.; validation, M.-H.Y. and R.K.; formal analysis, M.-H.Y.; investigation, M.-H.Y.; resources, M.-H.Y.; data curation, J.P.; writing—original draft preparation, M.-H.Y.; writing—review and editing, M.-H.Y., J.P. and R.K.; visualization, M.-H.Y.; supervision, M.-H.Y.; project administration, R.K.; funding acquisition, R.K. All authors have read and agreed to the published version of the manuscript.

**Funding:** This work was supported by a subaward from the Guam Coastal Zone Management Grant [NA18NOS4190202]; FEMA-4398-DR-GU Hazard Mitigation Grant Program [HMGP DR-4398-05].

**Informed Consent Statement:** Not applicable.

**Data Availability Statement:** Datasets related to this article can be found at https://data.mendeley.com/drafts/77s5dw57dx, an open-source online data repository hosted at Mendeley Data [30].

**Conflicts of Interest:** The authors declare no conflict of interest.

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
