# Peer review of "Identifying Characteristics of Guam’s Extreme Rainfalls Prior to Climate Change Assessment"

_water, doi:10.3390/w14101578_

Round 1
Reviewer 1 Report
This paper addresses the characterization of rainfall IDF curves to be considered in flood analysis and other related hazards. This is a relevant issue considering that usually there is a lack of information needed for constructing the IDF curves.
It is an interesting work; however, deeper analyses would be necessary.
Main aspects:
- As it is an issue largely studied, more references should be included comparing those studies with the proposed methodology. I indicate some just as an example:
- Langousis, Andreas & Veneziano, Daniele. (2007). Intensity-duration-frequency curves from scaling representations of rainfall. Water Resources Research - WATER RESOUR RES. 430. 10.1029/2006WR005245.
- Sujeewa Malwila Herath, Priyantha Ranjan Sarukkalige & Van Thanh Van Nguyen (2016) A spatial temporal downscaling approach to development of IDF relations for Perth airport region in the context of climate change, Hydrological Sciences Journal, 61:11, 2061-2070, DOI: 10.1080/02626667.2015.1083103
- Van de Vyver, H. A multiscaling-based intensity–duration–frequency model for extreme precipitation. Hydrological Processes. 2018; 32: 1635– 1647. https://doi.org/10.1002/hyp.11516
- TAN-DANH NGUYEN, VAN-THANH-VAN NGUYEN, and PHILIPPE GACHON (2007) A SPATIAL–TEMPORAL DOWNSCALING APPROACH FOR CONSTRUCTION OF INTENSITY–DURATION–FREQUENCY CURVES IN CONSIDERATION OF GCM-BASED CLIMATE CHANGE SCENARIOS. Advances in Geosciences. Pp 11-21
- Galiatsatou, P.; Iliadis, C. Intensity-Duration-Frequency Curves at Ungauged Sites in a Changing Climate for Sustainable Stormwater Networks. Sustainability 2022, 14, 1229. https://doi.org/10.3390/su14031229
- Bougadis, J., & Adamowski, K. (2006). Scaling model of a rainfall intensity‐duration‐frequency relationship. Hydrological processes, 20(17), 3747-3757.
- Related with the previous comment, Conclusion section should bring out the improvements or advantages that the method proposed in this paper provides regarding previous studies.
- In the Data section, information regarding seven rain stations is indicated. Rain series start between 2007 and 2012. However, authors mentioned in Conclusion section that the series for the seven rain stations are ranged from 2007 to 2021.
- Deeper analysis should be presented regarding the effect of the proposed method for flood studies and its applicability for other sites.
Minor aspects:
- Methodology section. Paragraph between lines 113 and 120 seems to have extra spaces between some words.
- Also, it should opportune to mention which non-parameter distribution was used to define the “observed” distribution (Gringorten?).
- Legend of Figure 6 should include symbol corresponding the observed data.
- References section. For some references, it is missed the site that they were “Retrieved from”. For example, USACE, 2020; WHO, 2015; etc.
Author Response
Please see the attached file: "Responses to Reviewers' comments"

Reviewer 2 Report
The manuscript is well written and easy to read. I have a few recommendations as below:
Line 30-31: Add reference/references
Add a flow chart figure that summarizes the steps in the methods.
Increase the quality of the figures (increase fonts and increase resolution (more than 300 dpi)
Line 180. Give exact numerical values
The manuscript can be accepted for publication after some minor revision.

Author Response

(The authors gave the same response as above.)
